# Cluster3D: A Dataset and Benchmark for Clustering Non-Categorical 3D CAD Models

**Siyuan Xiang** [1*]  **Chin Tseng** [1*]  **Congcong Wen** [1†]  **Deshana Desai** [1†]
**Yifeng Kou** [1†]  **Binil Starly** [2]  **Daniele Panozzo** [1‡]  **Chen Feng** [1‡]
[1]New York University   [2]North Carolina State University
https://cluster3d.github.io/

## Abstract

We introduce the first large-scale dataset and benchmark for non-categorical annotation and clustering of 3D CAD models. We use the geometric data of the ABC dataset, and we develop an interface to allow expert mechanical engineers to efficiently annotate pairwise CAD model similarities, which we use to evaluate the performance of seven baseline deep clustering methods. Our dataset contains a manually annotated subset of $22,968$ shapes, and $252,648$ annotations. Our dataset is the first to directly target deep clustering algorithms for geometric shapes, and we believe it will be an important building block to analyze and utilize the massive 3D shape collections that are starting to appear in deep geometric computing. Our results suggest that, differently from the already mature shape classification algorithms, deep clustering algorithms for 3D CAD models are in their infancy and there is much room for improving their performance.

## 1 Introduction

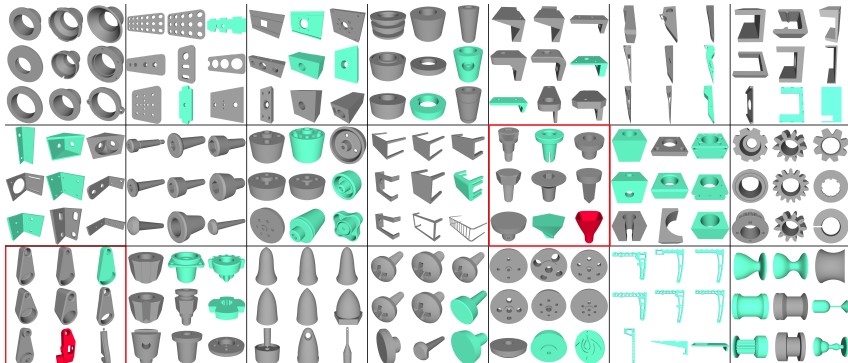

Figure 1: **The overview of Cluster3D** via a subset of clustering results from a baseline (DeepCluster), demonstrating the challenges of classification-based labeling in our task, due to many non-standard mechanical components (green). Each section shows some random CAD models in the same cluster. A red section shows a cluster with annotation violations highlighted at the red objects.

Shape classification is a core component in many modern 3D computer vision pipelines, and for which many datasets and benchmarks have been introduced in the last decade, usually focusing on a small number of object classes.

---

[*]Equal contributions.

[†]Equal contributions.

[‡]The corresponding authors are Chen Feng cfeng@nyu.edu and Daniele Panozzo panozzo@nyu.edu.

With the advent of large geometric collections of data, it is natural to expect that harder classification tasks will be solvable using modern data-driven classification approaches. However, we discovered that when the number of categorical classes becomes large and hierarchical, the task becomes very challenging. Even annotation of such object classes by human experts may yield varying classification labels, particularly because the necessary contextual information about the objects themselves are unavailable. We attempted to annotate the ABC dataset [35], which is composed of 1 million 3D CAD models, manually modeled by hobbyists and experts alike. The dataset has very complex and unbalanced class distributions, i.e., essentially being non-categorical. This makes the problem intractable even by our subject matter expert annotators (graduate students in mechanical engineering). *One may easily see such annotation challenges when trying to name each group of objects in Figure 1.*

Despite the challenges, without an automatic way to classify shapes, the practical utility of large geometric collections is hindered, especially considering that manual annotation is economically unfeasible at large scale, if not impossible to carry out due to incomplete information on the design intent of the original asset creator. Even when design information or engineering specifications are available, the content is in text form that cannot be easily related to features of the 3D model. The obvious alternative to supervised shape classification is the use of an unsupervised clustering method to group shapes, helping in both the annotation process and in the dataset exploration.

Surprisingly, we could not find any large-scale dataset or benchmark for deep clustering of collections of 3D shapes, especially non-categorical ones where clustering is more useful. To fill this gap, we propose to construct such a dataset based on the ABC dataset. Because of the non-categorical feature, we cannot annotate each object with a class label. Instead, we propose to annotate the pairwise 3D shape similarity relationship. Although it might sound even more intractable, we allow experts to focus on a small subset of carefully selected pairs (instead of all the pairs) to provide useful and scalable annotation. We developed a web-based user interface to implement this annotation workflow and to annotate $252,648$ selected pairwise similarities on $22,968$ ABC objects.

We benchmark seven clustering baseline methods to analyze the properties of our dataset. These clustering methods can be classified into two types: 1) two-stage clustering and 2) end-to-end deep clustering. For the two-stage clustering approach, we first perform deep representation learning on 3D mechanical components to extract high dimensional features, using pre-trained neural networks. Then we apply classic clustering algorithms, like KMeans [40], to group these learned features into different clusters. For end-to-end deep clustering methods, we combine representation learning, dimensionality reduction, and clustering in an end-to-end framework. Based on either the ground truth annotations or CAD model distance metrics, we respectively apply either external or internal cluster validation indexes [53] to evaluate their performances. Particularly, since we are the first to use similarity-based annotations, we need to design an external cluster validation index. We propose two formulations: one measuring the pairwise similarity accuracy, and the other measuring the compactness for the cluster elements.

We discovered that the performance of existing deep clustering methods is still insufficient for the automatic clustering of large datasets, and there is a lot of room for algorithmic improvement. We believe that our dataset will help by providing an objective metric on a large dataset specifically designed for this task. *We plan to continue collecting data and periodically update the dataset.* The dataset data, the annotation software, the implementation of all baseline methods, and scripts to run the evaluations are publicly released as open source using the MIT license.

In summary, our contributions are the following:

- To the best of our knowledge, Cluster3D is the first dataset focusing on non-categorical annotation for 3D mechanical components, which could stimulate a new direction for deep clustering on large-scale mechanical component collections.

- We propose a scalable and effective pairwise similarity annotation workflow, implemented in a graphical user interface, to allow experts to efficiently label a large number of object pairs (for a total of $252,648$ annotation pairs per annotator).

- We design/adapt 7 clustering methods on our dataset and benchmark their performances.

- We propose 2 external cluster evaluation indexes to evaluate the clustering results, using the similarity annotation. Also, we analyze our evaluation metrics, comparing them with several internal cluster evaluation indexes.

## 2 Related Work

We cover the related works more closely related to our main contributions: (1) large datasets of 3D models, (2) approaches for annotating 3D datasets, and (3) clustering algorithms and corresponding evaluation metrics.

**3D mechanical object datasets.** Large-scale 3D object datasets are routinely used for classification, instance segmentation, and shape reconstruction tasks [62, 43, 72, 10]. However, these datasets focus on a small group of categories, where each object can be uniquely and reliably be classified. Recently, large datasets of mechanical components have been introduced, which dramatically increase the difficulty in identifying and classifying 3D parts, due to their self-similarity and, generally, larger number of categories. Annotation of mechanical components requires more effort since the labeling work for such a dataset needs subject matter expertise rather than just common sense [34]. MCB [34] contains $58,696$ components and $68$ categories. The Fabwave dataset [2] contains $46$ classes of standardized part categories, with $4000$ variations under each of the standard part classes. Smaller datasets have been introduced in AAD [6] and ESB [31]. These datasets have been constructed by selecting components from a certain number of classes.

The ABC dataset[35] is different, as it was obtained by scraping all the public data available from OnShape. It contains more than one million mechanical components, and a large proportion of these components are non-standard, which means that are not belonging to standardized categories. The combination of massive scale and non-standard components makes it challenging to build a taxonomy on the dataset, even for our expert annotators. The number of classes and objects in each class does not fully represent the diversity of standard and non-standard parts seen in the product design category. In certain object categories, specific annotation labels do not depend on the shape alone but also its dimension and eventual intended application. For example, a fastener such as a washer and a gasket may look exactly the same, but the label categories vary based on the material specification and dimensions of the part. Annotation based on shape alone can lead to erroneous labeling and can often confuse annotators particularly when the true design intent of the user is not known or when the product assembly context is unknown.

Our Cluster3D dataset tackles this challenge directly, recasting the classification problem as an unsupervised clustering problem. To the best of our knowledge, our dataset is the first specifically designed for training and evaluating deep clustering methods on non-categorical 3D shapes.

**Interfaces for 3D models annotation.** Web-based platforms are commonly used for annotation acquisition since they require no front-end installation by annotators and the cloud infrastructure can support large 3D model datasets. MCB [34] developed a web-based platform with 3D viewers to provide enough information of 3D objects for annotation. ShapeNet [10] and PartNet [43] present web-based interface allows operating on 3D models and hierarchical 3D part annotation. The focus of these interfaces is segmentation and classification. In this work, *we introduce a web interface that enables efficient large-scale similarity annotation tailored for non-categorical datasets.*

**Unsupervised 3D representation learning.** Hand-crafted 3D descriptors has been studied as geometry-based methods [56], view-based methods[45, 61, 20, 11], or hybrid methods [37]. For 3D deep representation learning, these methods can be classified into point cloud-based method, view-based method, and volume-based method, depending on the different input data formats. Unlike supervised 3D deep learning that requires class labels [50, 51, 39, 70, 66], self-supervised methods are more suitable in our context. For example, Foldingnet [76], Atlasnet [24], TearingNet [47], and [1, 80, 12] are a series of work investigating the autoencoder architecture to learn the latent representation of point clouds. Rendered images from different views could also be used to learn 3D shape representations [54, 22, 26]. VConv-DAE [60] uses an autoencoder to learn the latent representation of 3D objects with voxel as input. A 3D shape descriptor network was also proposed to model volumetric represented objects[73]. Any of these descriptors, both hand-crafted and learned, can be used with the deep clustering methods discussed next.

**Deep clustering methods.** Deep clustering adopts deep neural networks to learn clustering-friendly representations [42] by integrating representation learning and clustering into an end-to-end model. The optimizing objectives are the network parameters and the clustering results. For a deep neural network, autoencoder-based models are widely used. DEC [28] is a classic deep clustering method: it first pre-trains the autoencoder with network loss for a few epochs, then fine-tunes the encoder network by optimizing KL divergence. DBC [38] achieves better clustering results compared to

DEC using a convolutional autoencoder instead of a feed-forward autoencoder. For DCN [74], the autoencoder network is pre-trained first, then the autoencoder network is jointly optimized with the K-means clustering results. Other methods, like DCC [58] and DEPICT [21], differ from DCN in that they use different clustering loss functions. Generative model-based deep clustering, such as variational autoencoder based method [33] and generative adversarial network based method [13], have also been studied. SCAN [67] relies on a two-step process for representation learning and clustering respectively. [32] maximizes mutual information for the clustering. DeepCluster uses the initial clustering results as pseudo labels for supervised training [8].

All these methods have been designed for 2D images but are adaptable to work on 3D object dataset by changing the features they use. We adapt 7 of these methods [24, 66, 23, 28, 67, 32, 8] to work on 3D models, and benchmark them on our newly introduced dataset for the first time.

**Clustering evaluation metrics.** To evaluate the success of a clustering algorithm and to enable objective comparisons between different methods requires an evaluation metric. While the choice is more obvious for supervised approaches, the situation is more challenging for clustering, especially in our setting where the entities involved are 3D models, which lack a canonical representation.

An ideal clustering result should maximize the intercluster distance (*compactness*), and minimize the intracluster distance (*separability*) [53]. The evaluation metrics can be classified into two major classes: external validation indexes and internal validation indexes [25].

*External validation indexes* use previous knowledge about the data to evaluate the clustering results [25]. In the computer vision community, the most common indexes are unsupervised clustering accuracy [77], normalized mutual information [69, 71], and adjusted rand index [29, 68]. Unsupervised clustering accuracy is the equivalent of usual classification accuracy, with the difference that it requires a mapping function to find the best mapping between the cluster assignment output and ground truth labels. Normalized mutual information measures the mutual dependencies between the cluster assignment and ground truth labels. Adjusted rand index is the corrected-for-chance version of the rand index [52], which measures the similarity between two clustering by comparing the all data pairs in the two clustering dataset. Besides, F-measure [65], entropy [59], purity [57] and other metrics [64] can also be applied.

The annotation matrix in our dataset can serve as the previous knowledge of the data for external evaluation. However, most of these indexes are using node labels rather than edge labels. Therefore, besides using [3] as one of the evaluation metric in our dataset, we also propose another evaluation metric to understand and analyze the baseline results, inspired by the purity index [57].

*Internal validation indexes* use the information intrinsic to the data [53], avoiding the need for additional external information [79]. Internal criteria can be further divided into two research topics [48]: 1) measurement of the fit of the cluster assignment and the inherent structure of the data and 2) the stability of the clustering results [48]. To measure the fit between the cluster results and structure of the data, compactness and separability is evaluated by computing the distances between clustered samples using the Dunn [17], Davies-Bouldin [15], Calinski-Harabasz [7], silhouette coefficient [55], and many other indexes in the literature [41]. An in-depth study on the stability of the clustering results, we refer to [5, 44, 36].

In our dataset, by defining the distance as Chamfer distance [4] or Jaccard distance [18], we can use these internal evaluation indexes. Particularly, we choose to use silhouette coefficient [55]. This index does not require the cluster centroid, which is more appropriate for our clustering results.

# 3   Cluster3D Dataset

We view the Cluster3D dataset as an undirected complete graph $\mathcal{G}(\mathcal{V}, \mathcal{E})$. The node set $\mathcal{V}$ contains each 3D CAD model in Cluster3D as a node $v$. Naturally, an edge $e_{i,j} \in \{+1, -1, 0\}$ in the edge set $\mathcal{E}$ stores the similarity annotation of two 3D CAD models $v_i$ and $v_j$ mentioned in the introduction. The edge labels $+1, -1$, and $0$ respectively indicate similar, dissimilar, and unknown relationship between two nodes. Next, we first explain how we create the Cluster3D dataset, and then discuss several important design considerations.

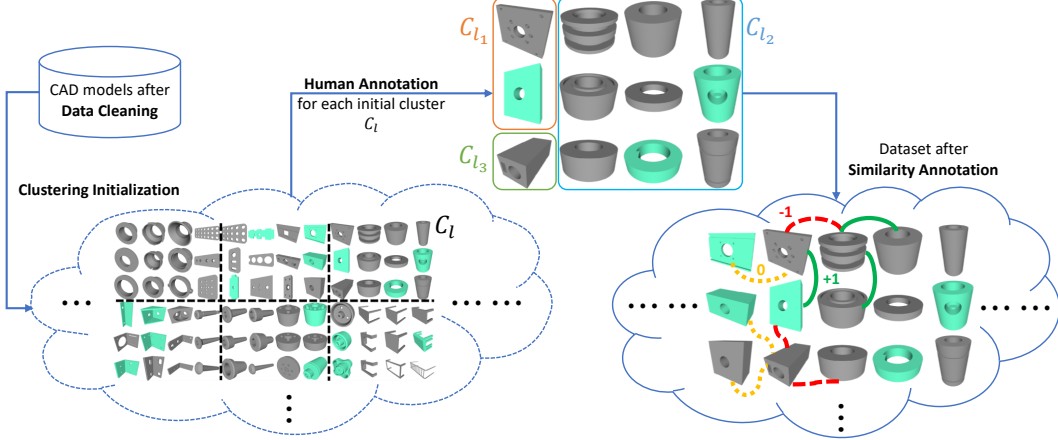

Figure 2: Cluster3D creation workflow.

## 3.1 Dataset creation workflow

We use the workflow illustrated in Figure 2 with the following major steps to create Cluster3D.

**Step 1: Data Cleaning.** We use the first four chunks of the ABC dataset [35]. We filter out all blank files and all files containing assemblies instead of single components, obtaining a dataset of $22,968$ CAD models.

**Step 2: Similarity Annotation.** Since 3D CAD models are non-categorical, it is challenging to assign object class labels for each node in Cluster3D. Instead, we propose to manually annotate a small set of carefully sampled edges storing the pairwise CAD model similarity. This novel scalable and efficient edge-based annotation is divided into two steps.

*Step 2.1: Cluster Initialization.* Before manual annotation, we first grouped all the CAD models into a set of initial clusters $\{\mathcal{C}_k | \cup_{\forall k} \mathcal{C}_k = \mathcal{V}, \mathcal{C}_k \cap \mathcal{C}_l = \varnothing \, \forall k \neq l, \text{and} \, |\mathcal{C}_k| \leq T, \forall k\}$, each containing no more than $T = 12$ CAD models, using a clustering method detailed in section 3.2. We then automatically assigned label 0 (meaning unknown) to all the edges across different clusters, i.e., $e_{i,j} = 0 \iff v_i \in \mathcal{C}_k, v_j \in \mathcal{C}_l, \text{and } k \neq l$.

*Step 2.2: Human Annotation.* We only manually annotate the edges residing inside the same initial cluster, i.e., $e_{i,j} \neq 0 \iff \exists k \in [1, K_I], v_i \in \mathcal{C}_k, v_j \in \mathcal{C}_k$, thus reducing considerably the human annotation cost. A number $A = 3$ of mechanical engineers served as our experts to provide their CAD model similarity annotations independently. For each one of the above initial clusters, e.g., $\mathcal{C}_k$, an annotator has to either *confirm* that CAD models inside $\mathcal{C}_k$ are all similar to each other, or further *divide* the cluster into smaller clusters until such confirmations can be made for each smaller clusters. The *confirmation* of a cluster $\mathcal{C}_l$ assigns all internal edges with the *positive* label $+1$, i.e., $e_{i,j} = +1 \iff v_i \in \mathcal{C}_l, v_j \in \mathcal{C}_l$. *Dividing* a cluster $\mathcal{C}_l$ into smaller clusters $\{\mathcal{C}_{l_t} | t \in \mathbb{Z}^+, \cup_{\forall t} \mathcal{C}_{l_t} = \mathcal{C}_l\}$ assigns all edges across those small clusters with the *negative* label $-1$, i.e., $e_{i,j} = -1 \iff v_i \in \mathcal{C}_{l_t}, v_j \in \mathcal{C}_{l_s}, s \neq t$. We record each annotator independently, i.e., the $a$-th annotator's annotation forms a complete edge set $\mathcal{E}^a$ over the same node set.

## 3.2 Design Decisions on the Annotation Workflow

**Why manually annotate similarity?** Although it has been widely used in geometry processing and machine learning, the concept of similarity can be vague and ambiguous when applied to 3D CAD models in Cluster3D. To determine whether two 3D models are similar or not algorithmically, there are two main criteria: geometric distribution similarity [63, 27, 19, 46] and visual similarity [11]. Yet for human beings, the mechanism to determine the similarity between two CAD models is often based on unconscious background knowledge [75, 16], which might be different from the similarity judgment encoded in existing algorithms [75, 14]. Therefore, even with mathematically defined 3D object similarity metrics, acquiring large-scale human annotations for pairwise CAD object similarity is still important to capture the underlying background knowledge.

**Reasons for cluster initialization**. We acknowledge that the choice of cluster initialization could introduce bias in our data collection, as a different clustering method would lead to a different subset of annotated edges. However, we argue that it is unavoidable, due to the sheer size of the dataset: it is impossible for human experts to annotate all the similarity relationships between one CAD model and all the remaining CAD models, as the quadratic number of edges is intractable. It would take 16 years of annotation time, assuming a 1 second time to annotate each edge in Cluster3D.

The most obvious, and unbiased approach, to restrict the manual annotation to a subset of the edges would be to do random sampling. This is however not an option for Cluster3D, as the sampling would be too unbalanced, as most of the edges indicate dissimilarity between objects. We needed a strategy that would give us a more or less even split between similar and dissimilar edges so that we could use the annotations to evaluate clustering methods in a balanced way.

After experimenting with different approaches, we found a method that, in our dataset, leads to a reasonable 1 to 1 ratio of similar and dissimilar edges: we use a clustering algorithm to overcluster the dataset in small clusters of 12 objects, and then ask users to annotate all edges within each cluster. Overall, this approach allowed us to get a good distribution of edge labels while annotating only 0.5% of the entries in our similarity matrix, making the annotation problem tractable with our budget and resources.

**Details for cluster initialization**. For cluster initialization, we use the MVCNN [66] based method. We opted for this method as it is the only image-based clustering method in our baselines, and in this way, we can do a fair comparison of the remaining six methods that are all using a point cloud representation.

Specifically, we first generate 12 images for all $22,968$ CAD models in our dataset, following the original settings in [66]. We use all these $12 \times 22,968$ images to train a convolutional auto-encoder network. Then the trained encoder is used to extract features for all these images. For each CAD model, we concatenate the twelve latent vectors from its corresponding 12 images to represent its features. Finally, $22,968$ features representing all the CAD models are clustered by KMeans algorithm. With the $K$ number in KMeans setting to be $2,000$, we have two thousand initial clusters for the human annotators. We continue to split the clusters with more than 12 models in the class using KMeans, until the contained number of models is not greater than 12. These clustered CAD models can be loaded into our database for annotation.

**Annotation interface.** We developed a web-based annotation application. The interface shows CAD models of one cluster at a time: it shows the 12 CAD models with checkbox, and initially all 12 checkbox are set to be checked. The annotators manually unmark the models which are considered dissimilar from the others, effectively annotating all edges linking the 12 models in the cluster. After confirmation, a new set of 12 models is shown.

**Conflicts in annotations.** We use one single annotated similarity matrix as the final outcome of our annotation procedure. In case of conflicts between different annotators, the majority wins. Note that for the final evaluation we also consider the individual similarity matrices of the different annotators.

**Data statistics.** Cluster3D has $22,968$ number of CAD models; therefore, totally there will be $22,968 \times 22,968$ number of similar or dissimilar edges between every two CAD models. Among them, $275,616$ edges are labeled by three human experts respectively. For the first annotator, $155,960$ edges are labeled as 1, representing these CAD model pairs are similar, and $119,656$ are labeled as $-1$, meaning these pairs are dissimilar. For the second and third annotator, they have labeled $130\,442, 145\,174$ similar edges, and $205\,582, 70\,034$ dissimilar edges respectively. We also check the consistency of the three annotators' labeling. Here consistency means the three annotator's label for a specific edge is the same. The total number of consistent label is $172,554$, occupying $62.6\%$ of the labeled edges. In the next release, we will double the number A and we expect to see higher consistency among the annotations.

# 4 Cluster3D Benchmark

## 4.1 Baseline methods

We adapt seven baseline methods to establish a benchmark for clustering algorithms. We divide these baseline methods into two types: 1) two-stage clustering, and 2) end-to-end deep clustering.

Two-stage clustering methods use a deep neural network to extract features for all CAD models, then apply a traditional clustering algorithm, such as KMeans. End-to-end deep clustering baseline methods integrate feature extraction and clustering in one framework: during the training process both network loss and clustering loss are minimized. Note that all these methods are considered as partitional clustering [9], i.e. one CAD model will only fall into one cluster.

Since some of the baseline methods are designed for 2D images, we adapt them for 3D CAD models. We use either point cloud or multi-view images as the representation format, and select suitable deep neural networks. A detailed description of all networks can be found in the supplementary.

**Two-stage clustering.**

*MVCNN:* We describe this algorithm in Section 3.

*AtlasNet:* To compute cluster of 3D point clouds using Atlasnet, we follow the original auto-encoder architecture to reconstruct 3D point cloud for each input 3D CAD object, and then predict latent vectors based on the encoder of the trained model. The CAD objects are clustered by using KMeans on the obtained latent features.

*BYOL:* BYOL is proposed to compute self-supervised image representation learning. We replace the image encoder (ResNet) with a point cloud encoder (PointNet) to learn a representation of a 3D CAD shape. We then apply the KMeans algorithm on the learned latent representation to cluster CAD objects.

**End-to-end deep clustering.**

*DEC:* To adapt the DEC algorithm, we initialize DEC with the AtlasNet architecture to auto-encode 3D point clouds as the input data. The deep auto-encoder is trained to minimize Chamfer loss and learns representations of the 3D shapes. We then follow the DEC algorithm by discarding the decoder layers and use the encoder layers as the initial mapping between the data and feature space. This is followed by joint optimization of the cluster centers and encoder parameters using SGD with momentum.

*DeepCluster:* We replace the convolution networks trained by the DeepClustering algorithm to use PointNet instead for encoding the point cloud data to predict cluster assignments. The algorithm is followed by alternating between clustering of the point cloud feature descriptors using K-Means and training the PointNet network using the multinomial logistic loss function.

*IIC:* Instead of the original IIC method for unsupervised image semantic task, we first randomly transform a CAD model to a pair of point clouds, and use PointNet as encoder to maximize mutual information between the class assignments of each pair. The trained model directly outputs class labels for each 3D CAD model.

*SCAN:* We adjust the pretext stage: Instead of using noise contrastive estimation (NCE) to determine the nearest neighbors, we use the auto-encoder of AtlasNet we have trained to output the feature vectors to generate the nearest neighbors set.

## 4.2 Evaluation Metrics

As discussed in Section 2, external and internal indexes are used for evaluating clustering results.

**External validation indexes.** We evaluate the clustering results of the baseline methods using three external validation indexes: *pair-wise accuracy*, *intra-cluster purity*, and *inter-cluster purity*.

*Pair-wise accuracy.* It is natural to compare the pair-wise clustering results with the annotated similarity matrix, which is the evaluation metric in correlation clustering [3]. We note that the similarity matrix is an undirected graph $\mathcal{G}(\mathcal{V}, \mathcal{E})$ on $N$ nodes. Let $e_{ij}$ denote the label of the edge relationship between object $i$, $j$, and $e_{ij} = e_{ji}$. $E = \{e_{ij}\}$ denote all the edges. $G' = (V', E')$ is the subgraph of of $G$, which is only composed of the known labels. $E' = \{e_{ij} | e_{ij} = 1 \vee e_{ij} = -1, e_{ij} \in E\}$. For the clustering results obtained from the baseline methods, $\hat{e}_{ij}$ denote the clustered edge relationship between object $i$, $j$. If objects $i$, $j$ are grouped into the same cluster, we assume the two objects are similar, therefore $\hat{e}_{ij} = 1$. Otherwise $\hat{e}_{ij} = -1$. The pair-wise accuracy is defined as: acc = $\sum_{e_{ij} \in E'} \frac{|\hat{e}_{ij} - e_{ij}|}{2n(E')}$, where $n(E')$ is the number of elements in $E'$. The range of the pair-wise accuracy is $[0, 1]$.

*Intra-cluster purity.* The pair-wise accuracy might miss information, since the clustering performance can be also evaluated cluster-wise [78]. The concept of purity [57] in node-labeled clustering evaluation inspired us, as it is used to evaluate the extent at which one cluster contains one single class. Similarly, we propose intra-cluster purity to detect the false positive edges in each cluster. Intuitively, the intra-cluster purity metric measures the extent at which each cluster contains similar edges. Let $K$ denote the number of cluster, $S = \{s_1, s_2, ..., s_K\}$ denote the set of the number of labeled similar edges in each cluster, $T = \{t_1, t_2, ..., t_K\}$ denote the set of the number of all known edges in each cluster. $T' = \{t'_i | t_i \neq 0, t_i \in T\}$ denote the subset of $T$, where only those clusters with at least one labeled edge are considered. Intra-cluster purity is defined as: $\frac{1}{n(T')} \sum_{i=1}^{n(T')} \left| \frac{s_i}{t'_i} \right|$. The range of inter-cluster purity is $[0, 1]$; 0 means the worst clustering result, and 1 means the best clustering result.

**Internal validation indexes.** It is not meaningful to use cluster centroids in our case, since the features of the CAD models are in high dimension. Therefore, we opt for the silhouette coefficient [55] method, which is widely used and does not require cluster centroids. Based on its definition, we need to determine the distance between every two objects. In our dataset, the objects are 3D CAD models which can be represented as point cloud or voxel. Therefore, we choose to use Chamfer distance [4] and Jaccard distance [18] as two distances between every two CAD models.

# 5   Benchmark Results and Discussions

**Experiment settings.** All the baseline methods are implemented using PyTorch [49] and run on an NVIDIA GeForce GTX 1080 Ti GPU. For hyperparameter settings, we tune learning rate and batch size for each baseline method. The learning rates for MVCNN-based method, Atlasnet-based method, BYOL-based method, DEC, DeepCluster, IIC, and SCAN are $0.0001, 0.001, 0.0003, 0.00001, 0.05, 0.0003, 0.0001$ respectively. The batch sizes for these methods are $60, 11, 10, 128, 50, 10, 96$ respectively.

**Clustering results using external evaluation metrics.** Figure 3-(Pair-wise accuracy) shows the benchmark results on the Cluster3D dataset, using the *pair-wise accuracy* and *intra-cluster purity* as external evaluation metrics. Since we do not known a priori the number of clusters in ABC, we test our baseline methods on seven different number of $K$, from 32 to $2,000$, following exponential growth.

*All baseline methods perform well with respect to intra-cluster accuracy, but their pair-wise accuracy is much lower.* Figure 3-(Intra-cluster purity) shows that all baseline methods can achieve *intra-cluster accuracy* higher than $0.8$ for all $K$. Some of the baseline methods can even achieve accuracy higher than $0.9$ when $K$ is no less than $128$. However, for *pair-wise accuracy*(Figure 3-(pair-wise accuracy)) results, it reveals that all these deep neural networks do not obtain enough ability to group similar CAD models, with their accuracy lower than $0.7$.

*Sensitivity to the cluster number $K$.* Figure 3-(Pair-wise accuracy) shows that most of the baseline methods' performances decrease as $K$ increases. We hypothesize that it is due to the imbalance of the annotated similarity matrix. There are a total of $431,576$ edges labeled as similar, and $395,272$ edges labeled as dissimilar for the three annotators. Therefore, we think the annotated similarity matrix might be biased towards the clustering results which have more similar pair predictions. For each baseline method, if the number of clusters $K$ increases, the CAD models will be more separate, causing more dissimilar pair predictions.

Figure 3-(Intra-cluster purity) shows that larger number of $K$ increase most baseline methods' performances. We believe it is because of the definition of *intra-cluster accuracy*. When $K$ becomes larger, the CAD models will be grouped into more clusters, which causes each cluster to be more pure. By definition, the more pure the clusters are, the higher the *intra-cluster accuracy* will be.

*Surprisingly, end-to-end deep clustering methods do not outperform two-stage clustering methods.* As shown in Figure 3-(Pairwise accracy), there is no obvious evidence showing higher performances of end-to-end methods (DeepCluster, DEC, IIC, SCAN), compared to two-stage clustering methods (AtlasNet-based method, BYOL-based method). Therefore, we believe it is necessary to study how to take the advantage of the clustering loss when we are training a deep neural network.

*MVCNN-base method is the only image-based method, and it was used during annotation: we believe these are the reason why it behaves noticeably differently than the other methods based on point clouds.* Figure 3-(Intra-cluster purity) shows the performance of MVCNN-based method is significantly lower than all other methods. Also, the MVCNN-based method is not as sensitive to the $K$ value as other methods.

*It still requires effort to study why some baseline methods perform differently than the overall trend with other methods.* Although most of the baseline methods show the same trend when the number of $K$ increases, SCAN and IIC are different. For SCAN, the input $K$ is used as the maximum number of clusters. Indeed, the actual number of cluster is often smaller than $K$, and it might be the reason that SCAN performances differently.

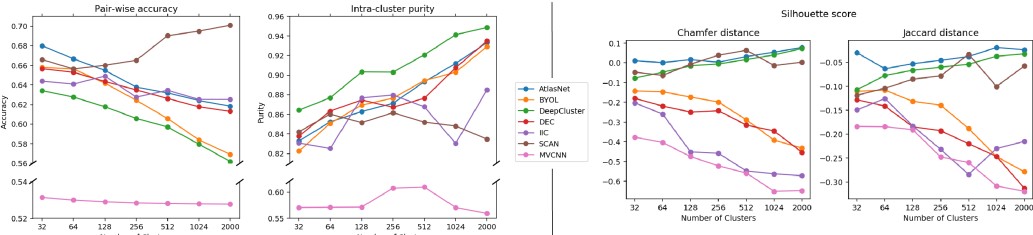

Figure 3: **Cluster3D benchmark results**.

**Clustering results using internal evaluation metric.** Figure 3-(Chamfer distance) shows the benchmark results using the *silhouette score* as internal evaluation metric. Using Chamfer distance and Jaccard distance lead to similar performances.

For all methods, we find that the clusters are not obvious different or even wrongly assigned, since most of the silhouette score is below $0$. Second, the AtlasNet-based method, DEC, SCAN methods perform better when the number of cluster $K$ increases, while other baseline methods show the opposite trend. Future investigation should be conducted to further understand this peculiar phenomenon.

## 5.1 Limitations and discussion

The major challenge in our study is the very high cost of annotating a similarity matrix which has a quadratic number of entries with respect to the number of objects in the dataset. We introduced a technique to reduce the annotation cost, but it is possible that the filtering introduced a bias in the annotations. This bias could be reduced by picking multiple initial clustering methods, which we plan to explore in the future.

The different evaluation metrics lead to different ranking for these baseline methods, suggesting that they evaluate different criteria. Identifying which metric is best for specific applications would be crucial to guide the development of clustering algorithms, and we believe it is an interesting venue for future work in deep clustering of 3D CAD models.

## 6 Conclusion

Cluster3D is a manually annotated dataset for the development and evaluation of clustering algorithms on 3D CAD models. We introduce the dataset, two external evaluation metrics based on the matrix, and benchmarked seven state-of-the-art clustering methods. Our conclusion is that the gap between human annotators and state-of-the-art methods is large: we believe our dataset will be an important resource to improve clustering methods for 3D geometry.

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
