# OpenReview forum: "Cluster3D: A Dataset and Benchmark for Clustering Non-Categorical 3D CAD Models"
_NeurIPS.cc/2021/Track/Datasets_and_Benchmarks/Round1 — Submitted to NeurIPS 2021 Datasets and Benchmarks Track (Round 1)_

### Official Review · Reviewer_WLaM · 2021-07-01
**A nice dataset, but several improvements could be made**

**Rating:** 4
**Confidence:** 4
**Clarity:** yes

**Strengths:**

1) the dataset is well motivated for benchmarking 3D shape non-categorical clustering algorithms. The task is important, and the dataset is the first dataset to benchmark the task.
2) the annotation is useful since several methods are benchmarked and interesting discoveries are obtained.
3) this data will inspire and attract more research on studying the task of 3D shape non-categorical clustering.

**Weaknesses:**

Since the clustering is usually subjective and ambiguous,
1) using a small pool of 3 annotators sounds too small to me. I'm concerned that there is a very big bias in the data. I suggest recruiting >10 annotators and provide some cross-validation for the bias estimates.
2) can you elaborate what are the criteria to rate the similarity between two shapes? for the same pair, using criteria A may rate them as similar but using criteria B may rate them dissimilar. How do you address this intrinsic ambiguity?
3) why only using three labels +1/-1/0? why not using floating numbers? maybe a pair (A, B) is more similar to (A, C) but both are given +1?

For the initial clustering and the strategy of only annotating intra-cluster pairs
4) why do you trust the initial clustering outcome?
5) will using different clustering algorithms give very different results?
6) why not using a collection of clustering algorithms, to reduce some bias?
7) if you think annotating all pairs is too expensive and annotating randomly sampled pairs is too biased towards negative outcomes, can you use a collection of clustering algorithms to suggest positive/negative ratings and sample equally from there?
8) if there is no inter-cluster annotation at all, I would guess the annotated dataset is much biased towards measuring the similarity between similar pairs, while mostly ignoring the pairs that are dissimilar.

For the benchmarks and experiments
9) How did you train the MVCNN? on what task/using what loss?
10) Can you show some qualitative diagnostic figures to compare/analyze the results? I only see one figure in supp, but more figures are needed.

**Additional Feedback:**

no

Post-rebuttal: I've read the other reviews and authors' rebuttal. However, I would like to keep my rating since I would prefer to see the dataset be well built before considering a publication. I think adding more annotators, more methods for clustering, more inter-cluster annotations, etc. are necessary to make this dataset/benchmark useful.

**Correctness:**

the claims are correct.
the dataset construction is mostly ok, but I have some concerns. (see questions 4-8 in the weakness section)
the benchmark, evaluation methods and experiments are good.

**Documentation:**

mostly ok, except no code release?

**Relation To Prior Work:**

yes

**Summary And Contributions:**

This paper, for the first time, proposes a non-categorical 3D CAD model similarity annotation dataset. The motivation is clearly stated and reasonable that many 3D mechanical part models are not easy to be given unambiguous labels. This paper enriches the existing ABC dataset with one more annotation: for 22K ABC models, sub-sampled 252K pairs are annotated with +1/-1/0 labels for their similarity/dissimilarity/unknown relationships. The authors used this dataset to benchmark seven 3D shape clustering algorithms and introduce two kinds of evaluation metrics for the benchmarking.

---

### Official Review · Reviewer_MbGU · 2021-07-02
**A large-scale 3D CAD similarity dataset based on ABC, substantial data contribution, but the data and baseline analyses are relatively weak and the possible bias problem is worrying.**

**Rating:** 5
**Confidence:** 4
**Correctness:** Basically.
**Clarity:** Clear.

**Strengths:**

+ Substantial annotations of the CAD model similarity on a very large-scale ABC dataset.
+ Interesting benchmark-setting which is beyond the current 3D object segmentation and classification.
+ Many baselines and important metrics are adopted in the experiments.
+ Some interesting results are presented.
+ This dataset would be useful for future 3D object understanding studies.

**Weaknesses:**

- Lacking enough and inspiring discussions and insights about the experiments. Some interesting results are shown but are not analyzed considering the dataset itself and different representations.
- My major concern is the bias may be introduced in the pre-processing, which is also mentioned by the authors. I think this should be dug very carefully for such a large-scale benchmark. Hope the authors revise this part and do more statistical tests on the possible bias.
- The consistent labels ratio is only 62.6% between the 3 annotators, I was very worried about this. What are the main reasons? How to improve this?
- More data statistics and visualized tables/figures are essential to check the data and annotation properties and quality.

**Additional Feedback:**

1. Can we use some recent self-supervised learning methods to do some more tests to dig the 3D representation effectiveness fully?
2. L118: objects[73] --> objects [73]
3. Please add citations for the baselines.
4. For BYOL, how to operate the data augmentation?

Post-rebuttal:
Thanks for the response from the authors. Some of my concerns (related works, experiment details) are addressed, but some remain—for example, the possible bias from the preprocessing and the consistency of similarity consistency problem. So I think this work still needs a major revision and data work to improve the data quality and benchmark design, which the other reviewers also mention. So I stand on my rating.

**Documentation:**

The detail on data collection and organization, availability, and ethical and responsible use is sufficient.
The maintenance is unknown but the authors have claimed that they would keep improving the dataset.
URL is given and contains some details.
Reproducibility: code is provided.

**Ethics:**

N/A.

**Relation To Prior Work:**

Some related works are missing, for example:
1. CAD model clustering:
CAD models clustering with machine learning
Clustering Techniques for Databases of CAD Models
SVM-based Semantic Clustering and Retrieval of a 3D Model Database
Shape-based clustering for 3D CAD objects: A comparative study of the effectiveness
...

2. CAD dataset:
3D Object Detection and Viewpoint Estimation with a Deformable 3D Cuboid Model
DeepCAD: A Deep Generative Network for Computer-Aided Design Models
Joint Embedding of 3D Scan and CAD Objects
Fusion 360 Gallery: A Dataset and Environment for Programmatic CAD Construction from Human Design Sequences
See the Glass Half Full: Reasoning about Liquid Containers, their Volume, and Content
IFCNet: A Benchmark Dataset for IFC Entity Classification
...

3. 3d representation:
Volumetric and multi-view CNNs for object classification on 3d data
...

**Summary And Contributions:**

This work mainly introduces a benchmark for complex 3D CAD model comparison. Upon the ABC dataset, this paper provides substantial model similarities for lots of CADs. Several widely-used metrics and 3D representation baselines are adopted to analyze the similarity annotations.

Though I think the data contribution is enough, the subsequent experiments and analyses lack enough insights and inspirations for future studies. And in the data processing, the whole annotation is based on a pre-clustering relying on clustering hyperparameters greatly, thus the annotation bias is unknown.  But the authors have honestly discussed this which I appreciate.

Overall, I think this word is potential for complex 3D object learning/understanding and is much different from current classification or segmentation tasks. But many aspects of the current version need to be improved. Thus, I think it should be revised carefully before accepted as a NeurIPS paper.

---

### Official Review · Reviewer_yjkQ · 2021-07-05
**Useful contribution, some additional explanation required**

**Rating:** 8
**Confidence:** 3

**Strengths:**

The use of pairwise similarity annotations makes sense for heterogeneous data, and this dataset will certainly be useful for evaluating unsupervised learning methods such as deep clustering on 3D models. The evaluations that the authors perform are helpful for establishing baseline performance in this area. The annotation framework clearly allows for ongoing annotation, so it is easy to anticipate that the annotation set will continue to grow.

**Weaknesses:**

My primary concern is based on the authors' point that "Annotation based on shape alone [in the dataset] can lead to erroneous labeling." Are the baseline clustering methods they use for evaluation all based on shape alone? I can't quite tell from the description, but it seems so. If so, then aren't these methods guaranteed to have a performance ceiling due to the fact that they don't have access to other factors that are described as important and which seem to be available to the annotations such as the object material, dimensions, design intent, etc.?

Put another way, how meaningful is it to use these annotations to evaluate clustering methods that are based on shape alone, when these features are known to be not just indeterminate but misleading in some cases with respect to pairwise similarity? Or is the point here to see how much information *can* be recovered via unsupervised methods from shape alone, with the understanding that this will always be imperfect? I think more careful explication of what it means to evaluate different clustering methods with respect to these annotations is required.

**Additional Feedback:**

No additional comments.

**Clarity:**

The paper is generally clear and well written. The "Reasons for cluster initialization" section I found particularly clear and helpful, and it addressed some of the immediate questions I had about the procedure.

For readers with no knowledge of CAD models, it would helpful to provide a few examples of what sort of categories or classes are represented in these types of datasets.

Some other points that bear revision/clarification:
- What does it mean when they say "This makes the problem intractable"? Does this mean annotators can't agree on the labels, that they can't even think of labels, that they don't know what they're looking at, or something else?
- What does "non-standard" mean in "a large proportion of these components are non-standard"?
- "Compactness" and "separability" appear to be reversed.
- I have some hesitation about the term "non-categorical" to describe this data. On its face, it suggests that the data is simply not amenable to labels, but I understand the intended usage to be something like "not readily reducible to a small number of categories." I'm not even quite that's accurate, however; perhaps the data *could* be reduced to some small number of meaningful categories, but a finer level of granularity is desired. Perhaps this term is in use in some relevant subfield that I'm not aware of; in any case, it would helpful to expand briefly on what exactly this term means.

**Correctness:**

The claims are all correct to the best of my knowledge. I believe the benchmarks are all performed correctly, and appropriate subject to the one caveat described above.

**Documentation:**

I believe that all these points are sufficiently addressed, with the exception of a section on ethical and responsible use. Perhaps there are no substantial ethical or use considerations here, but in any case these should be noted somewhere.

**Ethics:**

I can't think of any ethical considerations that require further discussion.

**Relation To Prior Work:**

Yes, the context is very clear.

**Summary And Contributions:**

The authors develop an annotation framework based on pairwise (dis)similarity for 3D CAD models. Through this framework, they elicit a large set of annotations for an existing CAD model dataset. They use these annotations to evaluate the performance of seven deep clustering methods. They also propose some modifications to existing clustering evaluation metrics to accommodate this dataset.

This paper clearly represents a useful contribution to the space of datasets. The context and procedures are generally well described and well justified. My one concern, detailed below, is about whether the annotations are really quite appropriate as a source of ground truth for clustering methods based on object shape alone, but this is a point that may be easy for the authors to address.

[Final remarks based on authors' responses:] Thanks to the authors for their responses. The clarification that the annotators only had access to the 3D CAD models resolves some of my uncertainty about what types of tasks the dataset will be useful for evaluating. I am concerned as several other reviewers are about the relatively low inter-annotator agreement, and I agree that more annotations are required, ideally with higher agreement. I think this type of dataset is very useful but these baseline measures need to be strengthened.

---

### Decision · Program_Chairs · 2021-07-26

**Decision:**

Reject

**Comment:**

There are few critiques regarding bias, subjectivity, and size of the proposed dataset. However, as clearly explained by the authors there is some inherent uncertainly and tractability issues with the size of the dataset. Overall, the novelty of the data and associated challenges, the dataset can be a useful contribution to the community but the paper still needs some improvement to be ready for publication. Unfortunatelly it can not be accepted in its current for for round 1. We encourage authors to carefully improve current version of the paper and submit it to round 2..